# 3D Modeling of the Solidification Structure Evolution and of the Inter Layer/Track Voids Formation in Metallic Alloys Processed by Powder Bed Fusion Additive Manufacturing

**DOI:** 10.3390/ma15248885

**Published:** 2022-12-12

**Authors:** Laurentiu Nastac

**Affiliations:** Department of Metallurgical and Materials Engineering, The University of Alabama, P.O. Box 870202, Tuscaloosa, AL 35487, USA; lnastac@eng.ua.edu; Tel.: +1-205-348-4844

**Keywords:** modeling of additive manufacturing processes, Electron Beam Powder Bed Fusion, Laser Powder Bed Fusion, rapid solidification, Grain structure evolution, texture and grain morphology, porosity defects, IN178

## Abstract

A fully transient discrete-source 3D Additive Manufacturing (AM) process model was coupled with a 3D stochastic solidification structure model to simulate the grain structure evolution quickly and efficiently in metallic alloys processed through Electron Beam Powder Bed Fusion (EBPBF) and Laser Powder Bed Fusion (LPBF) processes. The stochastic model was adapted to rapid solidification conditions of multicomponent alloys processed via multi-layer multi-track AM processes. The capabilities of the coupled model include studying the effects of process parameters (power input, speed, beam shape) and part geometry on solidification conditions and their impact on the resulting solidification structure and on the formation of inter layer/track voids. The multi-scale model assumes that the complex combination of the crystallographic requirements, isomorphism, epitaxy, changing direction of the melt pool motion and thermal gradient direction will produce the observed texture and grain morphology. Thus, grain size, morphology, and crystallographic orientation can be assessed, and the model can assist in achieving better control of the solidification microstructures and to establish trends in the solidification behavior in AM components. The coupled model was previously validated against single-layer laser remelting IN625 experiments performed and analyzed at National Institute of Standards and Technology (NIST) using LPBF systems. In this study, the model was applied to predict the solidification structure and inter layer/track voids formation in IN718 alloys processed by LPBF processes. This 3D modeling approach can also be used to predict the solidification structure of Ti-based alloys processes by EBPBF.

## 1. Introduction

The aim of this study was to model quickly and efficiently the microstructure evolution as well as pores and densification issues of IN718 alloy during the laser powder bed fusion (LPBF). The effect of laser scanning speed and power on the formation of IN718 microstructure, keyhole porosity, interlayer voids, and lack of fusion defects were studied in [1,2,3,4,5]. The effect of hatch spacing on the amount of lack of fusion voids was studied in [5] for a 5-layers 5-hatches IN718 build processed by LPBF. It was found in [5] that for a laser power of 60 W and a laser scanning speed of 1 m/s, the optimum hatch spacing to eliminate lack of fusion voids is about 60 microns. A thermal-fluid model to predict the temperature fields in a multi-layer-multi track LPBF IN625 build is presented in [6]. The model in [6] assumes a flat top surface without free surface flow and therefore it cannot predict the melt pool shape accurately. It was shown in [6] that for a laser power of 300 W and a laser scanning speed of 1230 mm/s, the maximum hatch spacing to avoid lack of fusion voids is about 75 mm. A Cellular Automaton (CA) model to simulate the grain structure formation in IN718 processed by Directed Energy Deposition (DED) process is presented in [7]. This CA model accounts for epitaxial growth, but it does not account for both heterogenous nucleation (which can occur at the edges and top surface of the pools) and rapid solidification conditions (see Equation (1) below). The CA model in [7] was used to generate the database to train and test the physics-informed machine learning methodology presented in [8]. Thus, the neural network model could be used as an additional tool to the CA model to quickly estimate the grain structure formation for large geometries in AM processing [8].

The current multiscale model consists of coupling a macroscopic thermal model with a solidification microstructure model [9,10,11]. The macroscopic model is fully transient and assumes that the thermal conduction is the relevant transport mechanism within the melt pool. Though the buoyancy is typically insignificant inside the melt pool (since Richardson number [12], Ri < 0.1), the importance of modeling convective transport within the liquid phase, recoil pressure and Marangoni convection in shaping the melt pool flow is important [13]. In spite of these limitations, this model is useful when operating in the thermal conduction mode. It predicts the observed trends in melt pool size and aspect ratio and it can be calibrated (similar to the approach in [14]) to bring the model output into reasonable quantitative agreement with the experiments over the operating parameter ranges.

Single layer laser remelting IN625 benchmark experiments using LPBF equipment were performed and analyzed at the National Institute of Standards and Technology (NIST) [15]. These experimental results were used to validate the 3D coupled macro-micro model as shown in [11]. Then, the validated model was applied to study the effects of process parameters on the microstructure evolution and texture in AM processed IN718 alloy. Two major objectives of this study were to: (1) develop a multi-scale simulation tool to predict the solidification microstructure formation of superalloys in multi-layer multi-track PBF processes and (2) gain scientific knowledge of phenomena involved in the microstructure formation of alloys processed under rapid solidification conditions via PBF processes. A key element for the model to address is how deposition of multiple tracks within a layer and then subsequent layers affects the remnant microstructure since a significant amount of solidified material is remelted. The main innovation in this research study consists of developing and coupling between the 3D macro and 3D micro modeling approaches and of adapting the 3D stochastic micro-modeling methodology to AM-rapid solidification conditions that will allow predictions of the 3D solidification structure of multicomponent alloy parts processed via AM. Furthermore, it is considered that the complex combination of the crystallographic requirements, isomorphism, epitaxy, changing direction of the melt pool motion and thermal gradient direction produced the observed texture and grain morphology. The entire domain is simulated with this modeling approach and therefore there are no limitations/assumptions in accounting for rapid solidification kinetics/remelting/epitaxy/isomorphism/etc. as they could be in other recently proposed modeling approaches for simulating grain structure evolution during solidification of metallic alloys in AM [16,17,18].

Other capabilities of the developed model include the ability to analyze the effects of melt transients, typically observed in the PBF processes. Thus, having a 3D modeling tool would be very useful in interpreting these phenomena. This model can be used to predict the experimental observations and then applied to determine the influence of various process parameters including scan path on the formation of texture and grain morphology. This multiscale modeling tool can also be used to predict the microstructure evolution in other AM processes and alloy systems such as Electron Beam PBF processed Ti6Al4V alloy. The ultimate goal of this study is to assist in improving the microstructure and mechanical properties of the processed alloys via AM and therefore to understand how to develop materials for aerospace applications.

## 2. General Methodology and Procedure: Multi-Scale Modelling Approach

Figure 1 shows a schematic diagram of the coupling between the 3D thermal model and the 3D solidification structure stochastic mesoscopic code [9,10,11].

The “Extract” code shown in Figure 1 has been developed to make this coupling possible. The Extract code computes and transfers all the information needed for the micro model, including mushy-zone thermal gradients, cooling rates and local solidification times. The macro model performs thermal computations to obtain the pertinent details of the temperature fields in the LPBF melting and solidification process. This information is then passed to the micro code, which subsequently computes the solidification structure of the LPBF processed material. The thermal macro-model is a fully transient 3D model of the heat transfer problem in the deposited material. For multi-track and multi-layer simulations, the moving energy source follows the given laser scanning directions.

The developed 3D micro model is based on a stochastic mesoscopic approach and differs from the classical “Cellular Automata” technique [9] in that it uses thermal history results from a macroscopic deterministic model and efficiently computes the solidification structure evolution in the entire simulation macro-domain.

The stochastic model for simulating the grain structure evolution includes heterogeneous nucleation and growth kinetics, as well as the growth anisotropy and grain selection mechanisms (see the model details in [9,10,11]).

The required input data for stochastic calculations are provided by the macroscopic model and include: (i) local cooling rates calculated at the liquidus and solidus temperatures, (ii) time-dependent temperature gradients in the mushy zone also calculated at the liquidus and solidus temperatures, and (iii) local solidification start time and end time. Local cooling rates calculated at the liquidus temperature are used to compute the nucleation parameters. Local average cooling rates and time-dependent temperature gradients in the mushy zone are used to compute the grain growth parameters. During solidification of the AM processed components, at least two types of grain morphologies are encountered: equiaxed grains and columnar grains solidified under a variable *G/V* ratio, where *G* and *V* are the local temperature gradient and solid–liquid (S/L) interface velocity of the mushy region, respectively. All aforementioned morphologies as well as the columnar-to-equiaxed transition are driven more or less by the same solidification mechanism that is the nucleation and growth competition of various phases in the mushy region. The stochastic mesoscopic models for equiaxed and columnar grains solidified under a variable *G*/*V* ratio are described in [9,10,11].

The stochastic model described above is adapted for rapid solidification conditions and multicomponent alloys, such as IN718. The procedure for the modeling of the multicomponent alloy systems was initially developed in [19] for alloys solidifying under normal solidification conditions. At normal cooling rates, the tip radius of the dendrite decreases as the solidification velocity increases [20]. However, as the cooling rate increases in the rapid solidification range, the tip radius increases, which is accompanied by a decrease in branching, and the equiaxed dendrite becomes globular/cellular. To account for the rapid solidification conditions (where solidification velocities typically exceed 0.1 m/s), the partition coefficient (*k**) and liquidus slope ($m_{L}^{*}$) of each alloying element needs be corrected using Aziz [21] and Baker and Kahn [22] equations, respectively:(1)$$k^{*}\left(V\right)=\frac{k_{e}+\delta_{i\;}V/D_{i}}{1+\delta_{i}\;V/D_{i}}\;\;\;\;\;\;\;\;m_{L}^{*}\left(V\right)=\frac{m_{L}}{1-k_{e}}\left[1-k^{*}\left(1-\ln\frac{k^{*}}{k_{e}}\right)\right]$$
where *k_e_* is the equilibrium partition coefficient, *V* is the S/L interface velocity, *D_i_* is the interfacial diffusivity, *δ_i_* is the atomic boundary layer thickness, and *m_L_* is the equilibrium liquidus slope. Experimental segregation testing can be used to determine the unknown *δ_i_/D_i_* ratio. The typical critical velocity value for solute trapping for metals is 5 m/s (using *D_i_* = 2.5 × 10^−9^ m^2^/s and *δ_i_* = 0.5 × 10^−9^ m) [20,21].

The coupled 3D simulation tool was applied to study the formation of grain structure in LPBF processed IN718 alloy. Basically, 3D simulations were performed in both the macro and micro codes. The extract code was then applied to couple these 2 simulation tools. The isomorphism/epitaxy between tracks/layers were accounted for in the simulations. 16-bit unsigned integers were used for unique grain ids, which also signify the crystallographic preferential orientation.

Model Validation. The coupled model was successfully validated in [11] against NIST experiments [15] for the IN625 laser remelting single-layer case in terms of melt pool dimensions and solidification grain structure formation (morphology and crystallographic orientation).

## 3. Results and Discussion

Table 1 shows the thermo-physical and material kinetics properties of IN718 alloy used in the current simulations, where T represents temperature in K ranging from 298 K to 2100 K, *ρ* is the density, *c_p_* is the specific heat, *K* is the thermal conductivity, *L* is the latent heat of solidification, *D_L_* is the liquid diffusivity, *Γ* is the Gibbs–Thomson coefficient, $T_{L}$ is the liquidus temperature; $T_{S}$ is the solidus temperature, $N_{0}$ is the nucleation density of the columnar grains, and the subscripts S and L denote solidus and liquidus, respectively. Specific heat and density of the Ar gas infiltrated powder bed/layer (which consists of two phases, IN718 power and Ar gas) are calculated by the volume weighted average method using a packing density factor that increases linearly with the fraction of solid (*f_S_*) from 0.55 [23] at *f_S_* = 0 to 1 for *f_S_* = 1. The specific heat and density of Ar as a function of temperature are taken from [24]. The thermal conductivity of the Ar gas infiltrated powder bed/layer (*K_pb_*) is calculated based on the data from [25,26] as *K_pb_ =* 0.016 *+* 6 × 10^−4^
*T*. *K_pb_* increased linearly with temperature from 0.19 W/m/K at 298K to 0.89 W/m/K at 1459 K. For temperatures greater than 1459K, *K_pb_* increases linearly with *f_S_* from 0.89 W/m/K at *f_S_* = 0 to *K_L_* at *f_S_* = 1.

Table 2 shows the process conditions used in the current simulation, where *P* is the laser power, *V_s_* is the laser scanning speed, Δ*t* is the time step used in the simulation, *A* is the absorption coefficient, *σ* is the laser beam radius, and $T_{i}$ is the initial temperature of the substrate and the processed powder material. The source term equation associated with the laser power is shown in Equation (1) in [11]. The length, width, and height of the current simulated geometry are 610 mm, 308 mm, and 244 mm, respectively. In the current simulations, 3 layers and 5 tracks per layer were considered and the laser scanning strategy was bi-directional between successive layers. The mesh size and number of cells for the macro-model is 2 μm and about 5.7 million cells, respectively, and the mesh size and number of cells for the micro-model is 1 μm and about 45.8 million cells, respectively.

The partition coefficient (*k^*^*) and liquidus slope ($m_{L}^{*}$) of IN718 alloy are calculated as a function of *V* (based on Equation (1)) in Figure 2. For this 3D geometry, the heat transfer boundary conditions (BCs) for the LPBF process are heat losses by conduction, convection, and radiation at the geometry top, edges, and bottom, and heat input due to the laser scanning source at the top of the geometry.

Figure 3 and Figure 4 show the simulation results for the 3 multi-layer 5 multi-track case at P = 188 W in ZY and ZX planes situated in the center of the simulated volume, respectively. Figure 5 and Figure 6 show the simulation results for the 3 multi-layer 5 multi-track case at P = 250 W in ZY and ZX planes situated in the center of the simulated volume, respectively.

The legend in Figure 3, Figure 4, Figure 5 and Figure 6 show the preferential crystallographic orientation angle of the columnar grains as 65535 color indices (*CI*). The orientation angles can be obtained from the legend using:(2)$$\theta =\frac{\pi }{2}\;\frac{CI}{65535}-\;\frac{\pi }{4}$$

Thus, when *CI*/65535 ratio varies from 0 to 1, ranges from −*π*/4 to *π*/4.

Lack of fusion and inter layer/track porosity defects can be clearly seen in Figure 3, Figure 4, Figure 5 and Figure 6.

The influence of laser power levels (P = 125, 188, 250 and 375 W) on the formation of grain size and orientation as well as of porosity defects is shown in Figure 7 and Figure 8. For the laser power of 188 W (Figure 3, Figure 4, Figure 7b and Figure 8b) the inter layer/track porosity defects were essentially eliminated. Lower and higher power levels (with respect to the P = 188 W) will increase the inter layer/track porosity defects. As shown in Figure 7 and Figure 8, the best oriented grains (grain colors/ids in the middle of the legends) with respect to the temperature gradient will typically grow to the top of the pool surface while lesser oriented grains will normally disappear. The process conditions selected for these simulations create cooling rates in the range of 2 × 10^4^–2 × 10^6^ K/s and temperature gradients in the range of 2 × 10^5^–2 × 10^7^ K/m. The average columnar grain size in Figure 7b is about 20 microns with a minimum grain size of about 5 microns and a maximum grain size of about 50 microns. As it can be seen in Figure 7b, the grain selection is quite strong: from about 50 grains at the bottom of the geometry, to about 30 grains at the middle of the geometry, and to about 20 grains toward the top of the geometry. Basically, about 2/5 of the bottom nucleated grains will survive in this 3-layer 5-track simulated AM geometry for P = 188 W. It can also be noted from Figure 7 and Figure 8 that the grain size is larger for P = 125 W as compared with all the other power levels, where the grain size is somewhat similar. Based on these simulations, the optimal power levels in terms of power efficiency, grain size and porosity reduction are the range of 175–200 W. This is in line with the experimental observations [1,2,3,4].

## 4. Computational Details

The micro-code was written in both CUDA C++ (GPU computing) and C++ (CPU computing). A sparse format was used for both input and output files. It was determined that:The CUDA 3D-Micro simulator is at least one order of magnitude faster than the CPU Micro-3D simulator using the Intel Skylake AL supercomputer;The CPU 3D-Micro simulator is fairly fast, typically taking about 1 h for a 3-multilayer 5-multitrack simulation using a 1 μm mesh size (46 million cells, RAM 8 GB);The CPU 3D-Micro simulator can run ~3 trillion cells on the Intel Skylake AL supercomputer (RAM 500 GB), which is equivalent of a 2.9 mm^3^ geometry using a 2 μm mesh size.

## 5. Concluding Remarks

An integrated 3D multiscale modeling approach was successfully developed by coupling a 3D transient thermal code and the 3D grain structure stochastic mesoscopic code. The coupled model was applied to simulate the microstructure evolution during multi-layer multi-track LPBF solidification of IN718 alloy. Thus, the simulation capability can be used to determine the effect of processing conditions on the inter layer/track porosity formation as well as on the grain size and orientation for AM processed IN718 alloys.

Based on the current simulations, it was determined that for a scanning speed of 800 mm/s, the optimal power levels in the term of power efficiency, grain size and porosity reduction are in the range of 175–200 W.

This modeling capability coupled with prior production experience can have a better insight into the effects of AM processing conditions on the formation of the IN718 microstructures and therefore, it would potentially reduce the number of experimental trials needed in developing a new AM practice.

The developed simulation capability can be applied to increase the fundamental understanding in the role of the complex combination of the crystallographic growth direction requirements, epitaxy (epitaxial growth), isomorphism (isomorphic growth), changing direction of the melt pool motion and thermal gradient direction on the formation of the solidification grain structure of metallic alloys.

## Figures and Tables

**Figure 1 materials-15-08885-f001:**
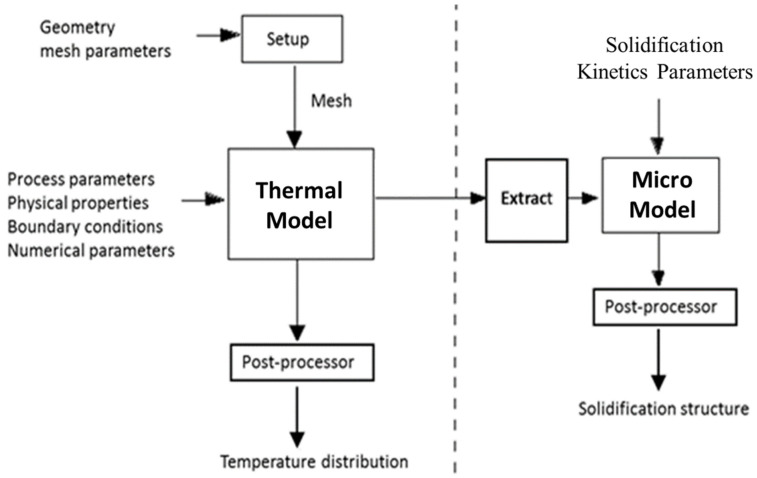
Diagram showing the coupling between the 3D thermal code and the 3D solidification structure evolution code.

**Figure 2 materials-15-08885-f002:**
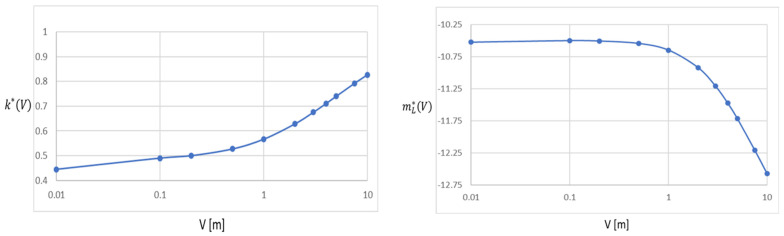
The effect of solidification velocity on the partition coefficient and liquidus slope for IN718.

**Figure 3 materials-15-08885-f003:**
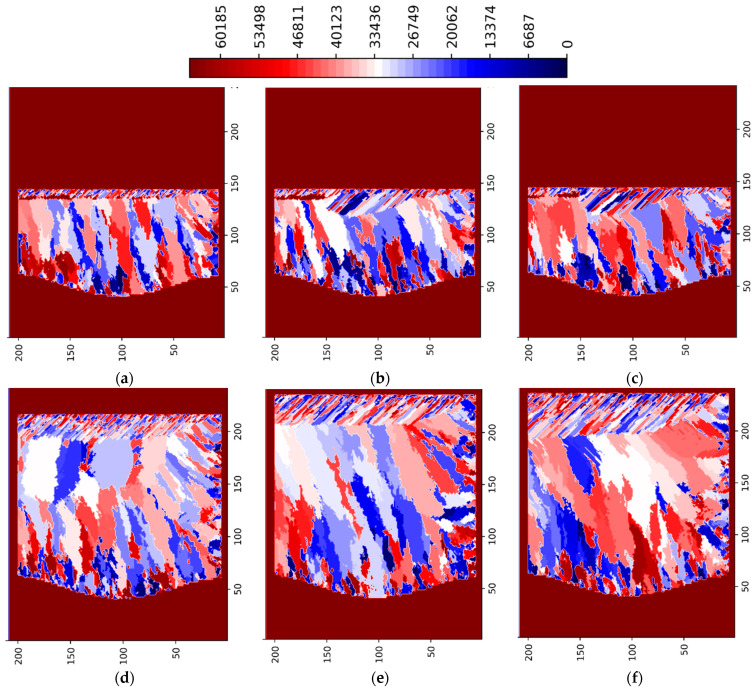
LPBF IN718 simulated grain structure (3 layers, 5 tracks)—ZY plane (P = 188 W): (**a**) layer 1 track 3; (**b**) layer 1 track 4; (**c**) layer 1 track 5; (**d**) layer 2 track 3; (**e**) layer 3 track 2; (**f**) layer 3 track 4.

**Figure 4 materials-15-08885-f004:**
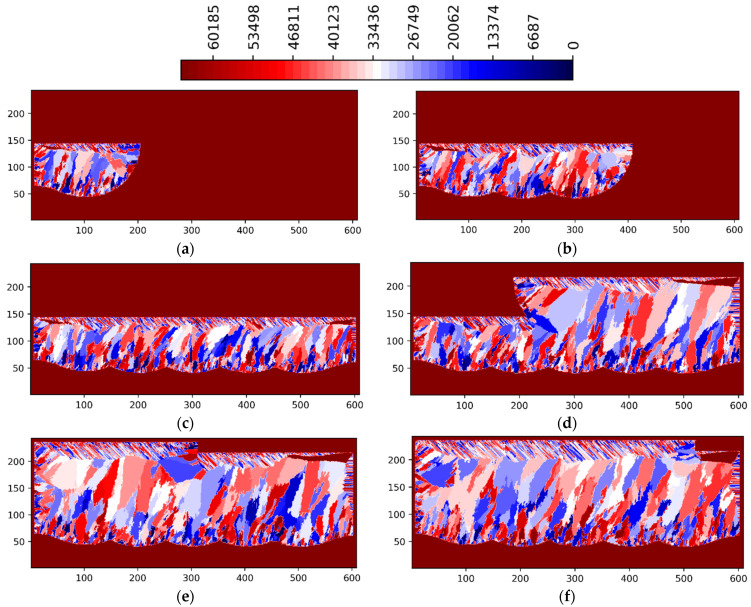
LPBF IN718 simulated grain structure (3 layers, 5 tracks)—ZX plane (P = 188 W): (**a**) layer 1 track 1; (**b**) layer 1 track 3; (**c**) layer 1 track 5; (**d**) layer 2 track 3; (**e**) layer 3 track 2; (**f**) layer 3 track 4.

**Figure 5 materials-15-08885-f005:**
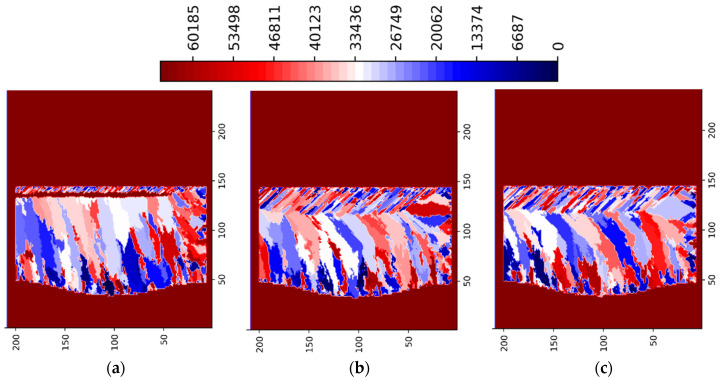
LPBF IN718 simulated grain structure (3 layers, 5 tracks)—ZY plane (P = 250 W) (**a**) layer 1 track 3; (**b**) layer 1 track 4; (**c**) layer 1 track 5; (**d**) layer 2 track 3; (**e**) layer 3 track 2; (**f**) layer 3 track 4.

**Figure 6 materials-15-08885-f006:**
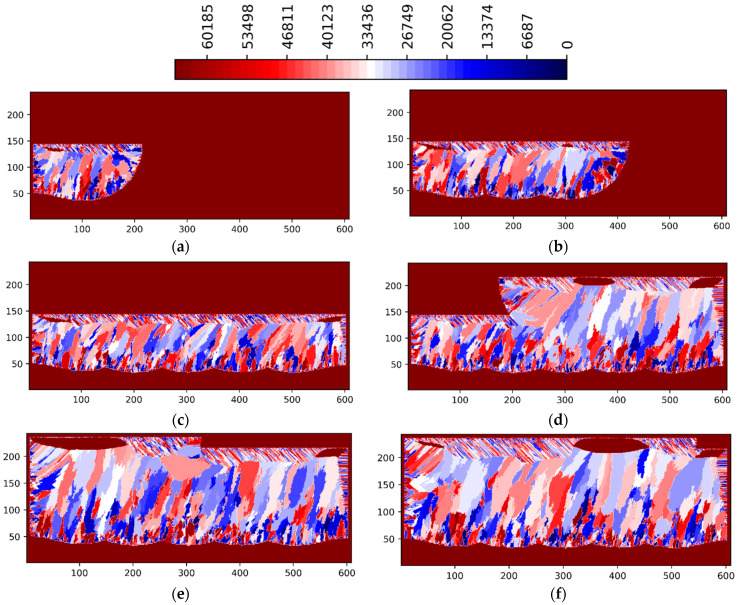
LPBF IN718 simulated grain structure (3 layers, 5 tracks)—ZX plane (P = 250 W): (**a**) layer 1 track 1; (**b**) layer 1 track 3; (**c**) layer 1 track 5; (**d**) layer 2 track 3; (**e**) layer 3 track 2; (**f**) layer 3 track 4.

**Figure 7 materials-15-08885-f007:**
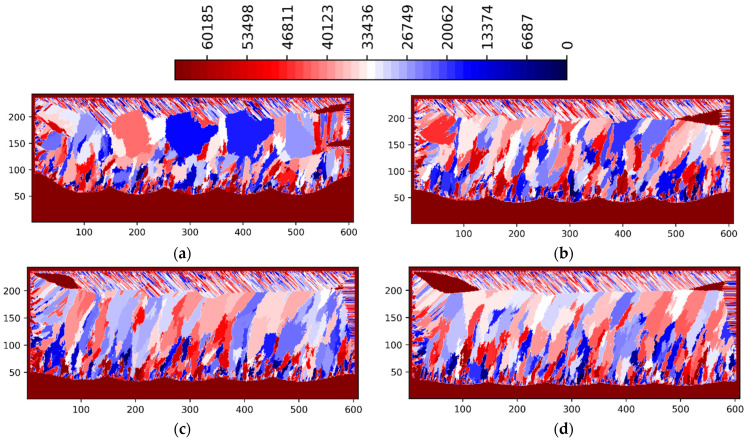
LPBF IN718 simulated grain structure (3 layers, 5 tracks)—ZX plane—layer 3 track 5: (**a**) P = 125 W; (**b**) P = 187.5 W; (**c**) P = 250 W; (**d**) P = 375 W.

**Figure 8 materials-15-08885-f008:**
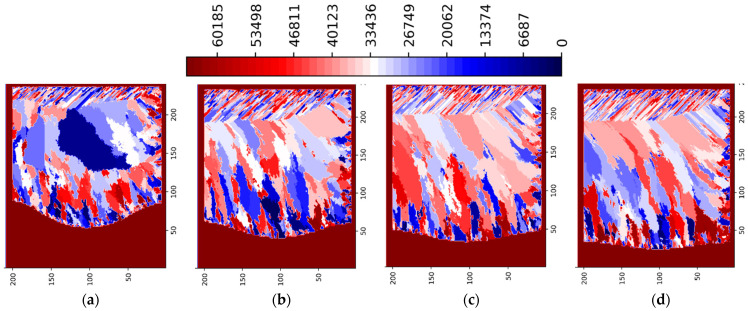
LPBF IN718 simulated grain structure (3 layers, 5 tracks)—ZY plane—layer 3 track 5: (**a**) P = 125 W; (**b**) P = 187.5 W; (**c**) P = 250 W; (**d**) P = 375 W.

**Table 1 materials-15-08885-t001:** Themo-physical and material kinetics properties of IN718 [9,10,27,28,29,30].

Property	Value	Unit
*ρ_S_*	8303.7 − 0.3622 T − 5 × 10^−5^ T^2^	kg/m^3^
*ρ_L_*	8366 − 0.488 T	kg/m^3^
*c_pS_*	305.72 + 0.3657 T − 8 × 10^−5^ T^2^	J/kg/K
*c_pL_*	720	J/kg/K
*K_S_*	3.8595 + 0.0208 T − 4 × 10^−6^ T^2^	W/m/K
*K_L_*	4.8985 + 0.0136 T	W/m/K
*L*	2.95 × 10^5^	J/kg
*m_L_*	−10.5	K/wt.%
*Γ*	3.65 × 10^−7^	K/m
*D_L_*	3.0 × 10^−9^	m^2^/s
*k_e_*	0.48	
$T_{L}$	1609.15	K
$T_{S}$	1459.15	K
$N_{0}$	1.0 × 10^10^	m^−2^

**Table 2 materials-15-08885-t002:** Process and material parameters used in the current simulation.

Parameter	Value	Unit
*P*	125, 188, 250 and 375	W
*V_s_*	0.8	m/s
*σ*	50	μm
*A*	0.3	-
Δ*t*	2.0 × 10^−5^	s
$T_{i}$	298.15	K

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
