# Peer review of "3D Modeling of the Solidification Structure Evolution and of the Inter Layer/Track Voids Formation in Metallic Alloys Processed by Powder Bed Fusion Additive Manufacturing"

_materials, 2022, doi:10.3390/ma15248885_

Round 1

Reviewer 1 Report

1) please check the capital letters of words before an acronym (example in the abstract, line 9: A fully transient discrete-source 3D Additive Manufacturing...

2) I feel that both title and keywords are too long: please try to reconsider them and be more concise.

3) In the Introduction, please describe clearly the fully dense and stochastic models you adopted in the methodology described in this research.

4) The reference list is too short: what about existing literature? Are there some published studies related to your research? If yes, what is your novelty compared to them?

5) In section 2, a poorly described thermal and micro model occurs. Please give more information. Regarding the 'extract code', please provide more details to make your research reproducible.

6) In the downloadable pdf, there are some issues with mathematic symbols: for example, on page 3, lines 107-111, some symbols are missing! Please check the entire manuscript about this aspect.

7) Section 3, page 4, line 138: the density symbol is or the rho greek letter?

8) Please provide more details about the case study in section 3.

Author Response

Dear Reviewer:
Thank you for your comments and suggestions to improve this paper. 
Please see my additional answers to your review below and the highlighted additions in yellow
color in the revision of the paper. Please also note that the meaning of the study is to simulate the 
microstructure evolution quickly and efficiently in multi-track multi-layer AM processed 
IN718 alloy and gain knowledge about the process parameters on the formation of the 
microstructure.
1) please checkthe capitallettersofwords beforeanacronym (example
in the abstract, line 9:A fully transient discrete-source 30 Additive
Manufacturing..
Corrected in the revised manuscript
2) Ifeel that both title and keywords are too long: please try to
reconsider them and be more concise.
3) In the Introduction, please describe clear1y the fully dense and 
stochastic models you adopted in the methodology described in this
research.
The multi-scale modeling approach is described in Section 2 and 
Refs. 9-11. 
4) The reference list is too short: what about existing literature? Are there
some published studiesrelated to your research? Ifyes, what isyour novelty
compared to them?
Thank you for your comment. I added 4 additional references with 
some discussion in the introduction section
5) In section 2, a poorly described thermal and micro model
occurs. Please give more information. Regarding the 'extract code', please
provide more details to make your research reproducible.
The multi-scale model was presented in more detail in refs. 9-11. 
As shown in lines 81-84, the Extract code shown in Fig. 1 computes and 
transfers all the information needed for the micro model, including mushyzone thermal gradients, cooling rates and local solidification times.
6) In the downloadable pdf, there are some issues with mathematic symbols:
for example, on page 3, lines 107-111, some symbols are missing! Please
check the entire manuscript about this aspect.
Corrected inthe revised manuscript
7) Section 3, page 4, line 138: the density symbol is r or
the rho greek letter?
Corrected in the revised manuscript
8) Please provide more details about the case study insection 3.
I have added an additional paragraph about the boundary conditions of the simulation study in 
lines 170-173. 
Also, the discussion of the simulation results shown at lines 174-180 and 223-241 describes in 
enough details the simulations presented in Figs. 5-10 including the key effects of the power 
level on the formation of the microstructure and porosity. 

Reviewer 2 Report

Microstructure evolution and voids formation are modeled in this work, together with the parametric study. The results are not novel, which have been reveled many published literatures aimed at numerical and experimental studies. Besides, the thermal modeling based on heat conduction cannot get high-fidelity results inside melt pool. Thus, I suggest to reject it. Followings are my comments for this manuscript, which could be considered in future revision.

1.      What are the details of your thermal model? Model geometry, governing equation, and boundary conditions should be given.

2.      Whether model is developed based on heat conduction, without the consideration of melt pool dynamics? If so, how to ensure the accuracy of the obtained temperature data? For example, the temperature gradient and cooling rate.

3.      From my knowledge, the temperature data calculated by thermal conduction presents large discrepancy, and cannot be used for microstructure prediction.

4.      More state-of-the-art papers focused on the modeling of microstructure and defects in PBF should be included in the part of literature review. Such as the words from Wing Kam Liu’s group in Northwestern University and T. Debroy’s group in Pennsylvania State University

5.      It is not reliable that only four papers are included in the Introduction Section.

6.      The author illustrates that the 3D thermal model is coupled with mico-model, where is the temperature field calculated by the thermal model?

7.      Experiments results in Fig.3 are obtained by laser remelting, not PBF.

8.      From the listed parameters in Table 1, it seems that thermal modeling is based on conduction mode. Thermal-physical parameters are temperature independent, and the melt pool dynamics is also ignored. Thus, the thermal results are not acceptable with larger error. Please see the following paper:DOI: 10.1007/s40192-021-00209-4

9.      Which is the x, y, and z direction in your proposed model? It should not be absence.

10.   The fusion lines of different lines should be labeled. It is difficult to understand now.

11.   This manuscript lacks insightful discussions of the simulated results.

Author Response

Dear Reviewer:

Thank you for your comments and suggestions to improve this paper.

Please see my additional answers to your review below and the highlighted additions in yellow color in the revision of the paper. Please also note that the meaning of the study is to simulate the microstructure evolution quickly and efficiently in multi-track multi-layer AM processed IN718 alloy and gain knowledge about the process parameters on the formation of the microstructure.

  1. What are the details of your thermal model? Model geometry, governing equation, and boundary conditions should be gi

The model geometry is described at line 163-165 in the revised manuscript.  As mentioned at line 162 in the manuscript, the governing equation including the source term associated with the laser power is described by Eq. (1) in Ref. [11]. 

Regarding boundary conditions, I have added the following sentence at lines 169-172: For this 3D geometry, the heat transfer boundary conditions (BCs) for the LPBF process are heat losses by conduction, convection, and radiation at the geometry top, edges, and bottom, and heat input due to the laser scanning source at the top of the geometry.

  1. Whether model is developed based on heat conduction, without the consideration of melt pool dynamics? If so, how to ensure the accuracy of the obtained temperature data? For example, the temperature gradient and cooling rate.

Thank you for this comment. The melt pool dynamics will be simulated with more accuracy in a future study. The purpose of the current multi-track multi-layer simulation tool is to model quickly and efficiently trends and gain knowledge about the formation of microstructure and porosity with respect to the laser power, speed, etc. for the entire AM processed geometry.  To model melt pool dynamics, free surface flow must also be included, which will improve accuracy, but it will also significantly increase the computational time.

  1. From my knowledge, the temperature data calculated by thennal conduction presents large discrepancy, and cannot be used for microstructure prediction.

Please see my answer for question 2. Also, as shown in my previous work (see Ref. [1]) and Fig. 3 in the current manuscript, the shape of the pool is not precisely described because the fluid flow is not considered; However, the overall simulated pool dimensions and morphology of the microstructure (type, size and shape) were in good agreement (see Fig. 3 in the manuscript).  I will include the fluid flow effects on the formation of the microstructure for multi-tack multi-layer setups in a future study.

  1. More state-of-the-art papers focused on the modeling of microstructure and defects in PBF should be included in the part of literature review. Such as the words from Wing Kam Liu's group in Northwestern University and T. Debroy's group in Pennsylvania State University

I have added 4 papers (see refs. 5-8] in the revised manuscript (3 from Liu’s group and 1 from Debroy’s group) .

  1. It is not reliable that only four papers are included in the Introduction Section.

4 more relevant papers (see refs. 5-8] were added in the revised manuscript.

  1. The author illustrates that the 30 thermal model is coupled with mico-model, where is the temperature field calculated by the thermal model?

The time-dependent temperature field is relatively trivial compared with the simulation of the microstructure evolution and it will not add any substantial information to the current paper.  

  1. Experiments results in Fig.3 are obtained by laser remelting, not PBF.

Thank you for this comment:  See corrections in the revised manuscript at lines 22-24 in the abstract and lines 54-56 in the Introduction section.  

  1. From the listed parameters in Table 1, it seems that thermal modeling is based on conduction mode. Thermal-physical parameters are temperature independent, and the melt pool dynamics is also ignored. Thus, the thermal results are not acceptable with larger error. Please see the following paper: DOI: 10.1007/s40192-021-00209-4

Thank you for this paper. I added as a reference in the current manuscript as Ref. 5. This paper is on multi layer-multi track LPBF of IN625.  Unfortunately, the paper does not show a comparison and errors between simulations with and without fluid flow but I will use the thermo-physical data shown in Table 1 of the “DOI: 10.1007/s40192-021-00209-4” reference in the future study. Also, the model in the paper assumes no free surface flo,: see page 9 in the paper: “With the assumption of the flat top surface in the model, the free surfaces of the melted tracks in the experiments are not considered in this study.”

As I answered questions 2 and 3, the objective of this study is to show that the fully transient coupled model is capable of modeling trends in the microstructure evolution in LBPF processes.  I will perform a sensitivity study regarding the thermo-physical properties of IN625 in a future study.   

  1. Which is the x, y, and z direction in your proposed model? It should not be absence.

Figs. 5-10 show the simulated cross-section planes, for example, in Fig. 5, ZY plane.  

  1. The fusion lines of different lines should be labeled. It is difficult to understand now.

The fusion lines can be observed from the microstructures shown in fig. 5-10, for example see Fig. 6d.

  1. This manuscript lacks insightful discussions of the simulated results.

The discussion of the simulation results shown at lines 174-180 and 223-241 is adequate.  It describes in enough details the simulations presented in Figs. 5-10 including the key effects of the power level on the formation of the microstructure and porosity.

Round 2

Reviewer 1 Report

Thanks to the changes applied to the manuscript, the overall quality of the research was highly improved. 

However, I still have minor concerns about the introduction section, where only a few contributions are cited to support the description of the results available in the existing literature. Indeed, the reference list is still very short, and 25% of references are self-citations. The final aim of this research is still not well described, and the novelty of the research is still not highlighted in the text.

Moreover, please check page 4, line 143, for missing math symbols.

There are also two sections numbered 3: 'Results and Discussion' and 'Computational details'.

Author Response

Dear Reviewer:

Thank you for your comments and suggestions to improve this paper.

Please see my additional answers to your review below and the highlighted additions in yellow color in the revision of the paper.

Please note that the meaning of the study is to simulate the microstructure evolution quickly and efficiently in multi-tracks multi-layers AM processed IN718 alloy and to gain knowledge about the process parameters on the formation of the microstructure. The entire domain is simulated at the micro-scale level. As mentioned in the manuscript, the 3D-Micro simulator is fairly fast, typically taking about 1 h for a 3-multilayer 5-multitrack simulation using a 1 μm mesh size (46 million cells, RAM 8 GB) on Intel Skylake Al supercomputer -

Thanks to the changes applied to the manuscript, the overall quality of the research was highly improved. 

However, I still have minor concerns about the introduction section, where only a few contributions are cited to support the description of the results available in the existing literature. Indeed, the reference list is still very short, and 25% of references are self-citations. The final aim of this research is still not well described, and the novelty of the research is still not highlighted in the text.

As shown in the revised version of the manuscript:

“The aim of this study was to model quickly and efficiently the microstructure evolution as well as pores and densification issues of IN718 alloy during the laser powder bed fusion (LPBF).”

To emphasize the novelty of this work, I have added the following paragraph in the revised manuscript:

“The main innovation in this research study consists of developing and coupling between the 3D macro and 3D micro modeling approaches and of adapting the 3D stochastic micro-modeling methodology to AM-rapid solidification conditions that will allow quantitative predictions of the 3D solidification structure of multicomponent alloy parts processed via AM.”

Regarding you “literature review” issue, I have added the following paragraphs and 6 more references in the revised version 2 of the manuscript as follows:

“The macroscopic model is fully transient and assumes that the thermal conduction is the relevant transport mechanism within the melt pool. Though the buoyancy is typically insignificant inside the melt pool (since Richardson number [12], Ri<0.1), the importance of modeling convective transport within the liquid phase, recoil pressure and Marangoni convection in shaping the melt pool flow is important [13]. In spite of these limitations, this model is useful when operating in the thermal conduction mode. It predicts the observed trends in melt pool size and aspect ratio and it can be calibrated (similar to the approach in [14]) to bring the model output into reasonable quantitative agreement with the experiments over the operating parameter ranges.”

“The entire domain is simulated with this modeling approach and therefore there are no limitations/assumptions in accounting for rapid solidification kinetics/ remelting/ epitaxy/ isomorphism/etc. as they could be in other recently proposed modeling approaches for simulating grain structure evolution during solidification of metallic alloys in AM [16-18]. “

  1. S. Turner, Buoyancy Effects in Fluids, Cambridge University Press, 1973, ISBN 978-0-521-08623-3
  2. A. Khairallah, A. T. Anderson, A. Rubenchik, W. E. King, Acta Materialia, 2016, vol. 18, 2016, 36-45.
  3. J. Goldak, A. Chakravarti, M. Bibby, Metallurgical Transactions B, 1984, volume 15, 299–305.

  1. M. Rolchigo, B. Stump, J. Belak and A. Plotkowski, Modelling Simul. Mater. Sci. Eng. 2020, Vol. 28, 065003.
  2. A. Koepf, M. R. Gotterbarm, M. Markl, C. Koerner, Acta Materialia, 2018, vol. 152, 119-126.
  3. M. Rodgers, J. D. Madison, V. Tikare, Computational Materials Science, 2017, vol. 135, 78–89.

Moreover, please check page 4, line 143, for missing math symbols.

Thank you for finding this typo; I corrected this in the revised version (see line 163)

There are also two sections numbered 3: 'Results and Discussion' and 'Computational details'.

Corrected  in the revised manuscript

Reviewer 2 Report

Most of the comments are not well addressed by the author in both the manuscript and responses to reviewer.

Additionally, there are also some evitable errors in the response letter, indicating the lack of attentions to deal with the reviewer’s comments.

Thus, it cannot be further considered and should be rejected.

In Q2, the modeling accuracy from heat conduction is not validated. Actually, the melt pool geometries can be calculated by Fourier equation, but the thermal behavior inside melt pool, including temperature gradient and cooling rate, cannot be accurately calculated by Fourier model. In other words, if you want to obtain the thermal data inside melt pool and use it to predict microstructure, dynamics flow must be considered.

In Q6, thermal data is the input for microstructure prediction, and generally thought as the dominant factor for the evolution of solidified microstructure. The thermal data must be contained in a work aimed at microstructure modeling in solidified melt pool.

In Q7, not only the experiment results in Fig.3, your research belongs to the field of laser remelting, not PBF.

In Q8, the thermal physical parameters in your heat conduction model are temperature independent, without the consideration of melt pool dynamics. Thus, it will lead to larger discrepancy and cannot be used for microstructures prediction.

Author Response

Dear Reviewer:

Thank you very much for your comments and suggestions to improve this paper.

Please see my additional clarification below and the highlighted additions in yellow color in the revision 2 of the paper.

Please note again that the meaning of the study is to simulate the microstructure evolution quickly and efficiently in multi-tracks multi-layers AM processed IN718 alloy and to gain knowledge about the process parameters on the formation of the microstructure. The entire domain is simulated at the micro-scale level. As mentioned in the manuscript, the 3D-Micro simulator is fairly fast, typically taking about 1 h for a 3-multilayer 5-multitrack simulation using a 1 μm mesh size (46 million cells, RAM 8 GB) on Intel Skylake Al supercomputer -

In Q2, the modeling accuracy from heat conduction is not validated. Actually, the melt pool geometries can be calculated by Fourier equation, but the thermal behavior inside melt pool, including temperature gradient and cooling rate, cannot be accurately calculated by Fourier model. In other words, if you want to obtain the thermal data inside melt pool and use it to predict microstructure, dynamics flow must be considered.

As far as I know, there is no paper on modeling free surface flow with recoil pressure and Marangoni convection coupled with microstructure simulations for multi-layer multi-track LBPF process.  The paper by Z. Gan et al (Ref. 6 in the manuscript) uses a thermal-fluid model but assumes a flat top surface of the melt pool without considering the free surface flow. 

And, as I mentioned in the previous review, I will add the fluid flow effects (including recoil pressure and Marangoni convection) on the melt pool in another study. 

I have added the following paragraphs in the revised version to clarify this:

“The macroscopic model is fully transient and assumes that the thermal conduction is the relevant transport mechanism within the melt pool. Though the buoyancy is typically insignificant inside the melt pool (since Richardson number [12], Ri<0.1), the importance of modeling convective transport within the liquid phase, recoil pressure and Marangoni convection in shaping the melt pool flow is important [13]. In spite of these limitations, this model is useful when operating in the thermal conduction mode. It predicts the observed trends in melt pool size and aspect ratio and it can be calibrated (similar to the approach in [14]) to bring the model output into reasonable quantitative agreement with the experiments over the operating parameter ranges. “

“The entire domain is simulated with this modeling approach and therefore there are no limitations/assumptions in accounting for rapid solidification kinetics/ remelting/ epitaxy/ isomorphism/etc. as they could be in other recently proposed modeling approaches for simulating grain structure evolution during solidification of metallic alloys in AM [16-18]. “

  1. S. Turner, Buoyancy Effects in Fluids, Cambridge University Press, 1973, ISBN 978-0-521-08623-3
  2. A. Khairallah, A. T. Anderson, A. Rubenchik, W. E. King, Acta Materialia, 2016, vol. 18, 2016, 36-45.
  3. J. Goldak, A. Chakravarti, M. Bibby, Metallurgical Transactions B, 1984, volume 15, 299–305.

  1. M. Rolchigo, B. Stump, J. Belak and A. Plotkowski, Modelling Simul. Mater. Sci. Eng. 2020, Vol. 28, 065003.
  2. A. Koepf, M. R. Gotterbarm, M. Markl, C. Koerner, Acta Materialia, 2018, vol. 152, 119-126.
  3. M. Rodgers, J. D. Madison, V. Tikare, Computational Materials Science, 2017, vol. 135, 78–89.

In Q6, thermal data is the input for microstructure prediction, and generally thought as the dominant factor for the evolution of solidified microstructure. The thermal data must be contained in a work aimed at microstructure modeling in solidified melt pool

See the answer for Q2 above.  Also, as mentioned in the paper, the purpose of the model is to simulate the grain structure evolution in the entire domain quickly and efficiently for metallic alloys processed through Electron Beam Powder Bed Fusion (EBPBF) and Laser Powder Bed Fusion (LPBF) processes.

In Q7, not only the experiment results in Fig.3, your research belongs to the field of laser remelting, not PBF.

I’m sorry but I don’t agree with your statement; as shown in this manuscript, the developed model can be used to simulate the multi-layers multi-tracks LPBF process.

In Q8, the thermal physical parameters in your heat conduction model are temperature independent, without the consideration of melt pool dynamics. Thus, it will lead to larger discrepancy and cannot be used for microstructures prediction.

The model can predict trends in the microstructure evolution.  Please see also my answer for this question above and in the previous review (question 8).

Round 3

Reviewer 2 Report

I had said many times that this manuscript cannot be further considered in current version.

a

Please be known that quick prediction is meaningful only when the modeling is performed with the consideration of key physical factors.

I emphasized the importance of modeling melt pool dynamics, not the free surface, because it had already been demonstrated that thermal data inside melt pool obtained from heat conduction is not reliable. The validation of geometry cannot make sure that the cooling rate and temperature gradient are well calibrated.

For the modeling of microstructure for multitrack multilayer printing considering both the melt pool dynamics and free surface, please see the following work. Yang et al. Phase-field modeling of grain evolutions in additive manufacturing from nucleation, growth, to coarsening

In your study, only the processing parameters of L-PBF are adopted and laser just irradiates the cubic substrate with flat surface. Please be known that L-PBF represents laser powder bed fusion. Where is the powder and where is the powder bed?

Last but most important, temperature independent properties are not acceptable for the modeling based on Fourier heat conduction.

In my view, the simulation must be performed again considering the variation of thermal physical properties versus temperature, otherwise, it cannot be further considered.

Author Response

Dear Reviewer:

Thank you very much for your comments and suggestions.

Please see my additional clarifications below and the highlighted additions in yellow color in the revision 3 of the paper.  Note that, as you requested, I have redone all the simulations in Figs. 4-9 using temperature dependent thermo-physical properties (see more details in my answers below).  

I had said many times that this manuscript cannot be further considered in current version.

Please be known that quick prediction is meaningful only when the modeling is performed with the consideration of key physical factors.

I emphasized the importance of modeling melt pool dynamics, not the free surface, because it had already been demonstrated that thermal data inside melt pool obtained from heat conduction is not reliable. The validation of geometry cannot make sure that the cooling rate and temperature gradient are well calibrated.

Please note that not only the pool dimensions but also the size and crystallographic orientation of the columnar structure were validated (see Fig. 2 in my paper (L. Nastac, 3D Modeling of the Solidification Structure Evolution of Superalloys in Powder Bed Fusion Additive Manufacturing Processes, Metals 2021, 11(12), 1995)

For the modeling of microstructure for multitrack multilayer printing considering both the melt pool dynamics and free surface, please see the following work. Yang et al. Phase-field modeling of grain evolutions in additive manufacturing from nucleation, growth, to coarsening

In my opinion, the phase field (PF) approach coupled with CFD models that consider free surface fluid flow is not yet feasible to be used in the simulation of the AM multi-layer multi track processes.  As shown in Yang’s paper, a commercial CFD package (Flow3D) was used that took 376 hours to simulated only 9 tracks and the PF model (~18 million) took 311 hours on NVIDIA Tesla M2090 GPU to simulate the microstructure for the same 9 tracks, total computational time is 687 hours (~29 days).  

The authors quoted at p. 9 “Besides, the present model can only simulate grain evolution in a limited number of layers and tracks due to the high computational cost.”

For comparison, the 3D-Micro simulator (see “Computational details” section in the manuscript) is fast, typically taking about 1 h for a 3-multilayer 5-multitrack simulation using a 1 μm mesh size (46 million cells, RAM 8 GB) on Intel Skylake Al supercomputer. The CPU 3D-Micro simulator can run ~3 trillion cells on the Intel Skylake AL supercomputer (RAM 500 GB), which is equivalent of a 2.9 mm3 geometry (e.g., many tracks/layers) using a 2 μm mesh size.

Note that I am very familiar with Flow3D modeling capabilities. The free surface model in Flow3D is sound but the CPU time it looks like (based on Yang’s paper) is unrealistic for simulating the LPBF process. This also shows that there is a lot of work to be done to improve the simulation speed for high fidelity macro-modeling of AM processes.  Therefore, a calibrated model (such as the proposed model in this manuscript) is useful for modeling the trends in the formation of the microstructure in AM

In your study, only the processing parameters of L-PBF are adopted and laser just irradiates the cubic substrate with flat surface. Please be known that L-PBF represents laser powder bed fusion. Where is the powder and where is the powder bed?

As mentioned in the manuscript, in the current simulations, 3 layers and 5 tracks per layer were considered and the laser scanning strategy was bi-directional between successive layers. Figs. 4-9 clearly show the formation of the microstructure in layers and track, which is LPBF.  The legends in Figs. 4-9 in the manuscript can only show the crystallographic orientation of the solidified grains.  Anything else (features as substrate, un-melted powder in the powder bed/layer, and top Ar region) in Figs. 4-9 is shown in a red color.  The simulation code cannot be easily changed to show all other features. Note that the powder bed/layer properties as a function of temperature are used in the current simulations (see details in the revised manuscript, lines 183-191).

Last but most important, temperature independent properties are not acceptable for the modeling based on Fourier heat conduction.

In my view, the simulation must be performed again considering the variation of thermal physical properties versus temperature, otherwise, it cannot be further considered.

I have redone all the simulations (see Figs. 4-9) using the thermo-physical properties as a function of temperature (including powder properties, lines 183-191) based on Refs. [23-30] in the revised manuscript (see also revised temperature-dependent properties in Table 1).
